# Molecular Analysis of SARS-CoV-2 Lineages in Armenia

**DOI:** 10.3390/v14051074

**Published:** 2022-05-17

**Authors:** Diana Avetyan, Siras Hakobyan, Maria Nikoghosyan, Lilit Ghukasyan, Gisane Khachatryan, Tamara Sirunyan, Nelli Muradyan, Roksana Zakharyan, Andranik Chavushyan, Varduhi Hayrapetyan, Anahit Hovhannisyan, Shah A. Mohamed Bakhash, Keith R. Jerome, Pavitra Roychoudhury, Alexander L. Greninger, Lyudmila Niazyan, Mher Davidyants, Gayane Melik-Andreasyan, Shushan Sargsyan, Lilit Nersisyan, Arsen Arakelyan

**Affiliations:** 1Laboratory of Human Genomics, Institute of Molecular Biology NAS RA, Yerevan 0014, Armenia; l_ghukasyan@mb.sci.am (L.G.); g_khachatryan@mb.sci.am (G.K.); t_sirunyan@mb.sci.am (T.S.); n_muradyan@mb.sci.am (N.M.); roksana.zakharyan@rau.am (R.Z.); a_chavushyan@mb.sci.am (A.C.); haivard@mail.ru (V.H.); 2Institute of Biomedicine and Pharmacy, Russian-Armenian University, Yerevan 0051, Armenia; m_nikoghosyan@mb.sci.am (M.N.); hovhannisyananahit19@gmail.com (A.H.); 3Bioinformatics Group, Institute of Molecular Biology NAS RA, Yerevan 0014, Armenia; s_hakobyan@mb.sci.am; 4Armenian Bioinformatics Institute, Yerevan 0014, Armenia; lilit.nersisyan@ki.se; 5Davidyants Laboratories, Yerevan 0054, Armenia; 6Laboratory of Evolutionary Genomics, Institute of Molecular Biology NAS RA, Yerevan 0014, Armenia; 7Department of Laboratory Medicine and Pathology, University of Washington, Seattle, WA 98102, USA; shahm4@uw.edu (S.A.M.B.); kjerome@fredhutch.org (K.R.J.); proychou@fredhutch.org (P.R.); agrening@uw.edu (A.L.G.); 8Vaccine and Infectious Disease Division, Fred Hutchinson Cancer Research Center, Seattle, WA 98109, USA; 9NORK Infection Clinical Hospital, MoH RA, Yerevan 0047, Armenia; lyudmila.niazyan@gmail.com (L.N.); davidyants@gmail.com (M.D.); 10National Center of Disease Control and Prevention, Ministry of Health RA, Yerevan 0025, Armenia; gayane.melik-andreasyan@ncdc.am (G.M.-A.); premier_h@yahoo.com (S.S.); 11SciLifeLab, Department of Microbiology, Tumor and Cell Biology, Karolinska Institutet, 17177 Solna, Sweden

**Keywords:** COVID-19, SARS-CoV-2, coronavirus, nanopore sequencing, Illumina sequencing, whole-genome sequencing, Armenia

## Abstract

The sequencing of SARS-CoV-2 provides essential information on viral evolution, transmission, and epidemiology. In this paper, we performed the whole-genome sequencing of SARS-CoV-2 using nanopore and Illumina sequencing to describe the circulation of the virus lineages in Armenia. The analysis of 145 full genomes identified six clades (19A, 20A, 20B, 20I, 21J, and 21K) and considerable intra-clade PANGO lineage diversity. Phylodynamic and transmission analysis allowed to attribute specific clades as well as infer their importation routes. Thus, the first two waves of positive case increase were caused by the 20B clade, the third peak caused by the 20I (Alpha), while the last two peaks were caused by the 21J (Delta) and 21K (Omicron) variants. The functional analyses of mutations in sequences largely affected epitopes associated with protective HLA loci and did not cause the loss of the signal in PCR tests targeting ORF1ab and N genes as confirmed by RT-PCR. We also compared the performance of nanopore and Illumina short-read sequencing and showed the utility of nanopore sequencing as an efficient and affordable alternative for large-scale molecular epidemiology research. Thus, our paper describes new data on the genomic diversity of SARS-CoV-2 variants in Armenia in the global context of the virus molecular genomic surveillance.

## 1. Introduction

Severe acute respiratory syndrome coronavirus-2 (SARS-CoV-2), which causes the novel coronavirus pneumonia COVID-19 [1], was first identified in China, in the city of Wuhan, in December 2019. The complete genome sequence of SARS-CoV-2 was published in January 2020 [2,3,4] and led to the development of real-time reverse transcription polymerase chain reaction (qRT-PCR) assays for SARS-CoV-2 detection that have served as a diagnostic standard during the ongoing COVID-19 pandemic [5]. Since then, whole-genome sequencing has been used for the evolutionary analysis of the virus, monitoring of circulating genetic lineages, and identifying signs of adaptation to hosts, which have important implications for treatment and vaccine development [6,7,8]. In the last two years, hundreds of studies were published describing country, region-specific and global insights into the dynamics and sources of SARS-CoV-2 importations and transmissions [9,10,11,12]. These results were obtained from the analysis of viral sequences, which were continuously sampled throughout the pandemic period.

In Armenia, the first confirmed case was reported on 1 March 2020. Since then, the number of positive cases reached 374,878 (as of February 2022) with several peaks at different time periods (Figure 1) with 8060 deaths and many re-infections [13]. In the absence of sequencing facilities in the country, virtually nothing was known about the transmission histories and epidemiological dynamics of the virus in Armenia. From March–August 2020, only three samples from Armenia obtained in July were sequenced in the Institute of Virology Charité Universitätsmedizin Berlin, which were deposited in the GISAID EpiCov [14,15] (accession ID: EPI_ISL_683449; EPI_ISL_683450; EPI_ISL_683451) in late December 2020. Another set of samples from September–November 2020 was sequenced by our colleagues at the Vaccine and Infectious Disease Division, Fred Hutchinson Cancer Research Center (USA), which became available in May 2021 (GISAID accession IDs are presented in Appendix A). This delay in analysis of molecular epidemiologic information hampered the informed and timely decision making by health authorities. In early 2021, our laboratory established the SARS-CoV-2 nanopore sequencing protocol and was able to perform almost real-time genomic surveillance by the monthly sequencing of viral samples during 2021 and 2022.

In the present study, we combine all above-mentioned genomic data to report the first molecular analysis of SARS-CoV-2 virus in Armenia in order to (1) understand the emergence and the transmission of the virus, (2) identify the most prevalent lineages at different time points, and (3) investigate the potential functional consequences of the mutations detected in the sequenced Armenian samples.

## 2. Materials and Methods

### 2.1. Samples

One hundred and ninety-one samples isolated from nasopharyngeal swabs were obtained from the Nork infection clinical hospital and the National Center for Disease Control and Prevention, Ministry of Health RA (NCDC), which served as primary testing sites. These samples were randomly selected from batches of COVID-19 positive samples tested at NCDC (Armenia), immediately after the confirmation of positive status, between June 2020–February 2022. Three additional samples previously sequenced at Charité Universitätsmedizin Berlin, Institute of Virology (Germany) and deposited in the GISAID EpiCov [14,15] were also included in this study (accessions: EPI_ISL_683451, EPI_ISL_683450, EPI_ISL_683449). The total number of samples was 194.

### 2.2. Real-Time PCR Detection of SARS-CoV-2

Automated RNA isolation was performed with Maxwell RSC Instrument using Maxwell RSC Viral Total Nucleic Acid Purification Kit (Promega Corporation Inc, Fitchburg, WI, USA). SARS-CoV-2 PCR testing was performed using Real-Time PCR Detection Kit for COVID-19 Coronavirus CE-IVD kit (Biotech & Biomedicine (Shenyang) Group Ltd., Shenyang, China) targeting ORF1ab and N genes. Samples were selected based on viral RNA load as measured by Ct values between 18–35 (Appendix A) for both targets.

### 2.3. Sequencing

Samples were sequenced with Oxford Nanopore and Illumina platforms (Appendix A). Nanopore sequencing of 146 samples was performed at the Institute of Molecular Biology NAS RA. Illumina sequencing of 45 samples was performed at Vaccine and Infectious Disease Division, Fred Hutchinson Cancer Research Center, Seattle, WA, USA. Five nanopore samples were additionally resequenced on the Illumina Nextseq platform to compare genome coverage and consensus level accuracy.

### 2.4. Nanopore Sequencing

Nanopore sequencing was performed according to “nCoV-2019 sequencing protocol v3 (LoCost) V.3” [16] based on ARTIC SARS-CoV-2 sequencing protocol with ARTIC nCoV-2019 V3 PCR panel [17,18].

#### 2.4.1. cDNA Generation

RNA samples were directly used for the first-strand synthesis using the LunaScript RT SuperMix Kit (New England Biolabs, Ipswich, MA, USA) with random hexamer and oligo-dT primers. Briefly, 8 μL RNA were mixed with 2 μL LunaScript RT SuperMix (5X) and were placed in a thermocycler and incubated for 2 min at 25 °C, followed by 10 min at 55 °C and 1 min at 95 °C and cooling to 4 °C. cDNAs were immediately used in subsequent steps.

#### 2.4.2. Amplicon Generation

Primer pairs from the ARTIC V3 primer scheme were used to amplify amplicons in cDNA [19]. Two multiplex PCR reactions were performed with 2.5 μL cDNA, 12.5 μL Q5 Hot Start High-Fidelity 2X Master Mix (New England Biolabs, USA), and 4 μL ARTIC V3 pool 1 (10 μM) or 4 μL ARTIC V3 pool 2 (10 μM). PCR cycling conditions were: 98 °C for 30 s followed by 35 cycles of 98 °C for 15 s, 65 °C for 5 min, and hold at 4 °C. The amplified products were purified with a 0.4x volume of AMPure XP beads (Beckman Coulter, Brea, CA, USA) to exclude small nonspecific fragments.

#### 2.4.3. Barcoding and Library Preparation

The purified PCR amplicons were treated with NEBNext End repair/dA-tailing Module (New England Biolabs, USA) and were barcoded with native barcodes and sequencing adapters (EXP-NBD104 and EXP-NBD114 kits Oxford Nanopore Technologies, Oxford, UK). Twelve or twenty-four samples were multiplexed in each sequencing run.

#### 2.4.4. Nanopore Sequencing

After priming the flow cell, 15 ng of the final sequencing library diluted to a final volume of 75 μL was loaded. Following the ligation sequencing kit (SQK-LSK109, Oxford Nanopore Technologies, UK) protocol, MinION Mk1B was used to perform genome sequencing in an FLO-MINSP6 R 9.4.1 flow cell for 3–6 h. The mean genome coverage across runs was 289 ± 189 (Appendix A Appendix A).

#### 2.4.5. Data Preprocessing, Demultiplexing, and Alignment

Base-calling and demultiplexing were performed using Guppy (v4.0.14). Raw FASTQ files were filtered and reads with lengths 400–700 b were selected using the ARTIC pipeline (release 1.1.0) [20]. Downstream analyses were performed using the nanopolish workflow implemented in the ARTIC pipeline [21]. The pipeline includes an alignment to the hCoV-19/Wuhan/WIV04 reference genome with minimap2 (2.17-r941) [22] followed by variant calling and consensus-building. Positions in consensus genomes with coverage lower than 20 were masked with “N” bases.

### 2.5. Illumina Short-Read Sequencing

The short-read sequencing of 50 samples was performed using the Illumina Nextseq platform following the protocol described in detail elsewhere [23]. In brief, sequencing libraries were prepared using the Swift Biosciences Normalase Amplicon protocol and SARS-CoV-2 amplicon panel, which contains 341 primer pairs spanning nucleotides 200 to 29,741 of the Wuhan reference genome (NC_045512v2) and produces amplicons ranging in length from 116 to 255 base pairs. The multiplex amplicon libraries were produced following the manufacturer’s recommendations, followed by 1.0× volumes of AMPure XP beads cleaning. Then, barcoded sequencing adapters were added to the amplicons. The resulting libraries were cleaned using 0.85× volumes of PEG NaCl and sequenced on the Illumina Nextseq instrument using 2 × 150 reads. Genomes were assembled using a custom pipeline described previously [23,24].

### 2.6. Phylogenetic and Variant Analysis

To perform preliminary QC of Armenian samples and select contextual samples, we initially screened our sequences with Nextclade [25] and PANGOLIN [26]. As contextual sample sequences GISAID nextregions selections were used (Global, Africa, Asia, Europe, North America, South America, and Oceania collections downloaded on 22 January 2022). In addition, we downloaded sequences for PANGO lineages B.1.1.163, B.1.1.419, B.1.1.528, and BA.1.1 that were detected in Armenian sequences, but were absent in the downloaded nextregions collections. In total, 17,721 contextual sequences were selected for combined analysis with Armenian samples.

Phylogenetic analysis, Nextstrain clade, and PANGO lineage identification was performed using the SARS-CoV-2 genomic epidemiology-specific pipeline implemented in Nextstrain 3.0.6 [25]. After preliminary QC (genome length more than 27,000, number of ambiguous reads < 3000), sequences were aligned to the reference genome (Wuhan/Hu-1/2019) with MAFFT v7.490 [27,28]. For phylogenetic context, we performed contextual subsampling based on genomic proximity [25] to select 10 sequences per country per year per month as representative sequences from the background. The final dataset consisted of 145 Armenian sequences (97 nanopore and 48 Illumina) and 9449 contextual background sequences. A maximum likelihood (ML) phylogenetic tree was constructed using IQ-TREE [29] under the GTR nucleotide substitution model. We used TreeTime [30] trait reconstruction on the resulting time-labeled tree to infer Armenia-centered trans-country transmissions with a “mugration model”. The temporal signal was evaluated by root-to-tip regression with TempEst v.1.5.3 [31].

The Coalescent Bayesian Skyline model of Armenian samples was constructed with BEAST v1.10.4 [32] with previously described parameters [9]. The model parameters were estimated with 40,000,000 Markov Chain Monte Carlo (MCMC) iterations, with 4,000,000 burn-in states and sampling every 1000 states. The MCMC parameter quality was assessed using Tracer v.1.7.1 [33] and were accepted if affective sampling size (ESS) values were higher than 100. The maximum clade credibility (MCC) tree was annotated using Tree Annotator v.1.8.4 [34].

### 2.7. Functional Annotation of SARS-CoV-2 Genomes

The functional annotation of SARS-CoV-2 genomes from Armenia included in the present study was performed using the Coronavirus Genome Analysis Tool (CorGAT) [35], where bioinformatic prediction of potential T-cell epitopes for SARS-CoV-2 were performed according to Kiyotani et al. (2020) [36].

## 3. Results and Discussion

### 3.1. Phylodynamic and Phylogeographic Analysis of Sequences

The total of 194 sequences represents 0.04% of 399,727 reported cases in Armenia as of 11 February 2022 (Figure 1).

Of the 194 sequenced samples, 145 (75%) sequences met the quality criteria (genome length of more than 27,000; number of ambiguous reads < 3000) and were included in the analyses. These 145 sequences represented 6 Nextstrain clades and 23 PANGO lineages (Figure 2A,B). The highest genomic diversity was noticed for the clades 21J (Delta) (nine PANGO lineages) and 20B (eight PANGO lineages). The analysis of root-to-tip regression with TempEst demonstrated a very strong temporal signal in our data (adjusted R^2^ = 1, 2.9 × 10^24^ on 1 and 143 DF, *p*-value: <2.2 × 10^−16^, Figure 2C).

The analysis of clades in sequencing samples and lineage-through-time plots (Figure 2D) indicated several rounds of clade substitutions within the time of sampling. June 2020–January 2021 samples were mostly represented with sequences belonging to the clade 20B with only two 19A sequences. Samples from March 2021 belonged exclusively to the clade 20I (Alpha), while May–July 2021 samples were in majority represented by 21J (Delta). Finally, late 2021–January 2022 were represented mostly by 21K (Omicron). Further analysis demonstrated inter-clade variability for the time of introduction, transmission routes, and PANGO lineages (Figure 3).

The clade 19A (B.4) was represented by only two sequences. The estimated time for their introduction was late July (21 June 2020, Date Confidence Interval: 9 June 2020–27 June 2020); however, this clade was not detected in later samples suggesting its replacement in the Summer of 2020. The analysis of the transmission routes indicated Iran as the source of introduction for these sequences around early March 2020, which corresponds well with the date of the first positive case of COVID-19 in Armenia identified in a traveler from Iran [37]. Thus, we can speculate that July was the period of substitution of the 19A clade with 20B, which fits with the global domination of clades harvesting Spike D614G [38].

The clade 20B formed three big clusters associated with different introduction events as well as a few single introductions that did not result in large intra-country transmissions. Early introduction sources were Italy (2 March 2020, Date Confidence Interval25 February 2020–5 March 2020) and New Zealand (20 April 2020, Date Confidence Interval: 15 March 2020–22 May 2020). Interestingly, the latest introduction (12 October 2020, Date Confidence Interval: 18 September 2020–9 November 2020) observed was almost exclusively represented by the B.1.1.163 lineage imported from Russia that formed a big intra-country cluster. The estimated time of importation coincided with the sharp increase in positive cases in September–November 2020 (Figure 1).

The variant of concern 20I (Alpha) had two introductions in Armenia. According to the temporal analysis, the lineage B.1.1.7 was introduced around 24 December 2020 (Date Confidence Interval: 26 September 2020–10 January 2021) and 24 November 2020 (Date Confidence Interval: 6 September 2020–15 January 2021). The Nextstrain pipeline identified Jordan and Germany as the main transmission route for the 20I (Alpha) clade sequences. The 20I (Alpha) was primarily responsible for the third peak of positive cases in late February–March 2021 (Figure 1). The introduction of this variant in Armenia happened with several months’ delay compared with the West European Countries and resulted in considerably fewer infections as well. One of the reasons can be the strict travel restrictions and a negative 72 h PCR test requirement for inbound travel [39]. The other reason can be the peak of infection caused by 20B (B.1.1.163) in late 2020.

The estimated earliest introduction of the clade 20J (Delta) in Armenia was 28 December 2021 from India (B.1.617.2). The majority of the 20J (Delta) sequences, however, were represented by the AY.122 lineage (39 of 56 sequences) forming a single cluster with an estimated date 22 February 2021 (Date Confidence Interval: 8 December 2020–17 March 2021), introduced from Liechtenstein. More recent sequences for this clade have diverse geography (Bahrain, Denmark, Greece, India, Jordan, Portugal, South Africa, Spain, Suriname, and Turkey), but mostly without producing many secondary cases according to the phylogenetic tree.

Finally, the sequences belonging to the 21K (Omicron) clade demonstrated the highest geographical diversity of introduction (Brazil, France, Maldives, Mexico, Netherlands, and Sweden). The earliest inferred date for this clade introduction was estimated at 6 December 2021 (Date Confidence Interval: 11 October 2021–29 December 2021).

Both 20J (Delta) and 21K (Omicron) caused a sharp increase in positive cases compared to previous waves. On the other hand, the deaths accompanying the 21 (Delta) wave were considerably higher compared with the 21K (Omicron) (Appendix A Appendix A), which is in line with observations in other countries [40,41].

Thus, our phylodynamic and phylogeographic analysis of the SARS-CoV-2 Armenian sequences allowed us to identify and characterize virus clades/lineages transmissions to Armenia. The results indicate multiple inter-country importations and their persistence in the country.

### 3.2. Functional Annotation of Variants

We performed the functional annotation of analyzed sequences using Nextclade [25] and CorGAT [35] tools (Appendix A Appendix A). Besides known effects of clade/lineage signature mutations (for the most comprehensive list see https://covariants.org/, accessed on 12 May 2022), we were interested in the functional consequences of “private mutations” (reversions to reference, mutations ascribed to different clades, and mutations that are for reversions or belonging to other clades) as defined by the Nextclade app.

The reversions were identified in six 20I (Alpha) and one 21K (Omicron) samples, which constituted 54% and 12% of clade sequences, respectively. Most private mutations ascribed to other clades were detected in 20B (27 sequences) and 21J (Delta) (12 sequences); however, only seven such mutations were found in more than one sequence. Finally, at least one private unlabeled mutation was identified in all the 20B, 20I (Alpha), 21J, and 21K (Omicron) sequences. Overall, 138 mutations were detected in more than two samples. The functional annotation of all mentioned mutation types is provided in Appendix A Appendix A. No specific enrichment for HLA epitopes [42,43,44], evolutionary selection, or secondary structure elements were observed in private mutations compared to all known mutations (HLA-epitopes *p* _Fisher exact_ = 1; Selection pressure *p* _Fisher exact_ = 0.45; Secondary structure *p* _Fisher exact_ = 0.63); however, the overlap between these categories was observed for some of the mutations (Figure 4).

HLA loci association with COVID-19 incidence, risk, or severity has become one of the research focus areas. Many associations have been reported in various countries and populations [43,45]; for example, HLA-A*02:01 was predicted to have a high binding to the virus epitopes and shown to be protective against COVID-19 severity, while HLA-A*01:01 was considered a risk factor for the disease [43]. Moreover, another recent study evaluated the association of HLA loci with the side effects of mRNA vaccines [46]. Recently, the study of HLA loci association with COVID-19 has also identified HLA-C*04:01 as a risk factor for severe disease in Armenia [47]. Previous population-scale studies identified the common HLA alleles in the Armenian population (HLA-A*02:01, HLA-A*01:01, HLA-A*24:02, HLA-A*03:01, HLA-B*51:01, HLA-B*35:01, and HLA-B*49:01) [48]. We evaluated the representation of epitopes targeted by these loci in our sequences. We identified epitopes for two protective HLA loci (HLA-A*02:01 and HLA-A*24:02) and three risk loci (HLA-A*01:01, HLA-A*03:01, and HLA-B*51:01). Our results demonstrate that the majority of sequences harbor mutations in epitopes with a high-binding affinity to protective HLA loci, while only a few sequences showed the presence of the mutations associated with low-binding HLA alleles (Table 1, Supplementary Material Appendix A). No HLA-C*04:01 locus-related epitopes/mutations were observed; however, this allele was not present in the CorGAT’s HLA annotation dataset. Thus, the question of whether mutations in viral sequences may be related to the observed association of HLA-C*04:01 with disease severity remains open.

Polymerase chain reaction (PCR) is the current standard method for COVID-19 clinical diagnosis from clinical samples. Therefore, we conducted a reassessment of published diagnostic PCR assays, including those recommended by the World Health Organization (WHO), through the evaluation of the possible effect of identified mutations on the efficacy of recommended primers and probes used for PCR detection of SARS-CoV-2 with the Nextclade app. In 143 sequences, we observed 39 mutations in the viral genome that did not match RT-PCR primers/probes for SARS-CoV-2 detection (Appendix A Appendix A). However, mutations located in template regions for US CDC N3 and China CDC Orf1AB primers and probes did not influence the primer binding since we obtained the N gene PCR signal in all studied samples (Appendix A Appendix A).

### 3.3. Comparison of Oxford Nanopore and Illumina Sequencing

In this study, 97 nanopore sequencing samples and 48 Illumina sequencing samples were included, which gave us an opportunity to evaluate the performance of both approaches. First, we assessed the number of missed nucleotides in the Nextclade app analysis, which can indicate gaps in genomes because of insufficient read coverage. Out of the 97 nanopore samples, 86 had missing sites, while in Illumina samples, they were detected only in 5 samples. The length distribution of missing sites was 146 ± 176 nt and 70 ± 22 nt in nanopore and Illumina samples, respectively. The large SD in nanopore samples is caused by a high number of single missing sites as well as supposed amplicon drop-outs (Figure 5).

We also sequenced five samples by two methods, so we compared them to evaluate the correspondence of clade/lineage assignment with nanopore and Illumina short-read sequencing. We compared the Nextstrain clade and PANGO lineage assignment for the consensus these sequences produced (Table 2, Appendix A Appendix A).

In three cases out of five, the clade and lineage were in agreement between nanopore and Illumina sequencing. IMB2-1/2021 isolate was initially assigned to B.1.1 with nanopore sequencing, while Illumina consensus was identified as B.1.1.7. The analysis of the BAM files for this strain indicated that the amino acid substitutions and deletions characteristic of B.1.1.7 lineage also existed in the nanopore sequencing reads; however, they did not pass the quality check during calling by nanopolish variant pipeline (Appendix A Appendix A). Moreover, the temporal analysis also indicated January as the estimated time for this lineage introduction in Armenia (see Section 3.1). In another case, the Illumina sequence for IMB1-5/2021 was assigned to B1, while nanopore sequencing assigned the same sequence to B.1.1.163. Overall, our data suggest that Illumina sequencing can produce better consensus sequences than nanopore; the possible reason could be differences in coverage generated in the two approaches and also specific amplicon dropouts described for the ARTIC primer scheme [16,49]. However, nanopore sequencing can serve as an efficient and affordable alternative to Illumina (short-read) next-generation sequencing and be used for the epidemiologic surveillance and molecular-genetic analyses of SARS-CoV-2. This is particularly important in countries with underdeveloped NGS sequencing facilities, such as Armenia.

## 4. Conclusions

Our study added new data to the global context of genomic epidemiology of SARS-CoV-2 and provided a holistic overview of the emergence, transmission, and diversity of the virus in Armenia. We identified multiple introductions of genomic lineages and their relations with the dynamics of positive cases during 2020–2022. Interestingly, the majority of importations inferred by phylogeographic analyses were through airway travels, while ground transportation played very little or no role, consistent with closed ground borders in Armenia and neighboring countries almost immediately after the first positive case in Armenia [50]. The majority of early importations were from countries with a considerably large Armenian diaspora (such as Russia and Kazakhstan) as well as touristic destinations (Italy) and much a wider geography for later VOC lineages.

The functional analysis of mutations (both lineage defining and private) identified a considerable number of mutations that affected the binding of predicted viral epitopes to protective HLA loci. Consistent with the previous reports, such mutations were present in the majority of VOC lineages compared to older lineages [51].

Our results also show multiple mutations in regions covered by several primers/probes compared with the reference sequence of the virus. This observation is of particular importance, since mutations may lead to the alteration of the sensitivity of qRT-PCR tests. Diagnostic tests mostly used in Armenia target ORF1ab and N genes, and our results suggest that identified mutations will not influence their accuracy.

The results of the study again emphasize the need for constant sequencing-based surveillance of SARS-CoV-2 strains for public health decision making and health care. Illumina short-read whole-genome sequencing platforms enable accurate sequence determination and are currently the method of choice for SARS-CoV-2 sequencing [52]. However, whole-genome sequencing essential for epidemiological monitoring and surveillance of viral pathogens is still challenging in many countries with limited technical resources. While the superiority of Illumina platforms over nanopore sequencing has been established in several studies [17], the latter still can serve as an efficient and affordable alternative to short-read next-generation sequencing and be used for epidemiologic surveillance and molecular-genetic analyses of SARS-CoV-2. This is particularly important in countries with underdeveloped NGS sequencing facilities, such as Armenia, and can play an important role in shaping local, national, and regional COVID-19 response strategies.

It is also worth noting the limitations of this study. First, the number of genomes sequenced and analyzed was small compared to the total number of positive cases in the country. Moreover, the sample collection and sequencing started in Autumn 2020, which limited our ability to accurately reconstruct events close to the first dates of the epidemic. The geography of trajectories for lineage importations also should be treated with caution since they were inferred from the phylogenetic analyses based on the limited number of background sequences selected. Unfortunately, we did not have access to the travel and contact history information, which definitely would otherwise improve the accuracy of the results.

However, even with these limitations, we believe that this paper is an important contribution in an attempt to fill the knowledge gap and demonstrate the importance of the real-time genomic surveillance of SARS-CoV-2 for informed and timely public health interventions.

## Figures and Tables

**Figure 1 viruses-14-01074-f001:**
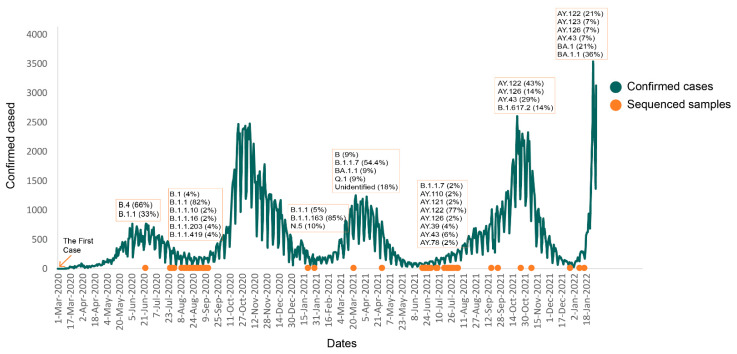
Daily incidence of reported confirmed cases for Armenia, sampling dates, and clade distribution of the sequenced samples.

**Figure 2 viruses-14-01074-f002:**
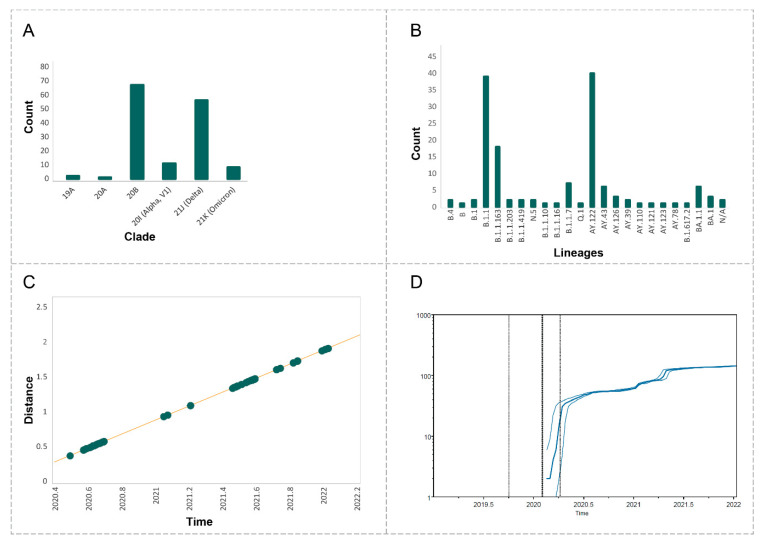
(**A**) Distribution of SARS-CoV-2 lineages according to Nextstrain clade nomenclature. (**B**) Distribution of SARS-CoV-2 lineages according to PANGO lineages nomenclature. (**C**) Root-to-tip regression analyses using TempEst. The plot represents linear regression of root-to-tip genetic distance within the ML phylogeny against sampling time. (**D**) “Lineage through time” plot of the phylogenetic tree of Armenian samples. The solid thick line represents the log median of ancestral lineages present at each time interval. Thin lines represent 95% confidence intervals for the number of lineages.

**Figure 3 viruses-14-01074-f003:**
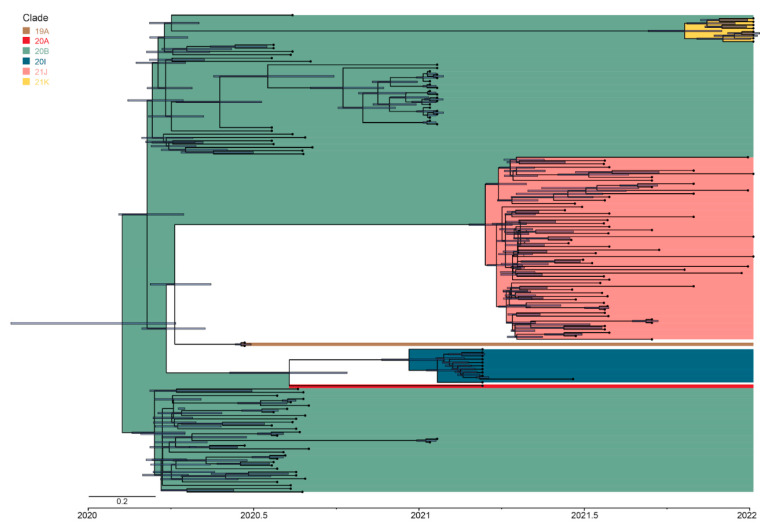
Annotated MCC tree of Armenian sequences. Background colors indicate Nextstrain clades; tip labels indicate PANGO lineages. Bars at nodes represent 95% height posterior density.

**Figure 4 viruses-14-01074-f004:**
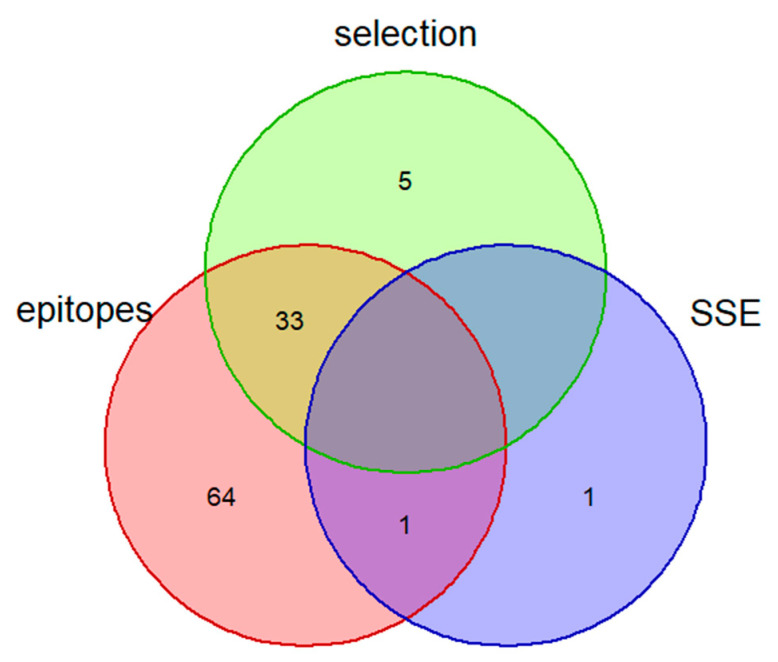
Venn diagram depicting the overlap between the Nextclade private mutations in HLA epitopes, under evolutionary selection and causing changes in secondary structure elements (SSE) observed in the Armenian samples (n = 145).

**Figure 5 viruses-14-01074-f005:**
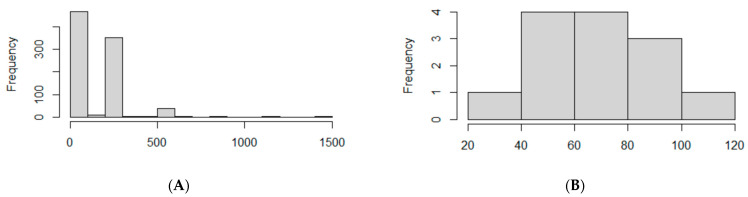
The bar graph represents the size distribution of missing sites for (**A**) nanopore and (**B**) Illumina samples. The *X*-axis represents the size of missing sites; *Y*-axis represents the absolute frequency (number) of missing sites per size.

**Table 1 viruses-14-01074-t001:** Absolute count of sequences harboring private mutations in HLA epitopes across clades.

Clade	HLA-A*02:01, HLA-A*24:02 (Protective Alleles)	HLA-A*01:01, HLA-A*03:01, HLA-B*51:01 (Risk Alleles)
19A	3 mutations	2 mutations
20B	228 mutations	128 mutations
20I (Alpha, V1)	68 mutations	24 mutations
21J (Delta)	453 mutations	303 mutations
21K (Omicron)	76 mutations	19 mutations

**Table 2 viruses-14-01074-t002:** Comparison of the PANGO lineage and GISAID clade assignment for consensus sequences produced with Oxford Nanopore Technologies and Illumina.

Sample	Oxford Nanopore		Illumina	
	PANGO Lineage	Nextstrain Clade	PANGO Lineage	Nextstrain Clade
IMB1-1/2021	B.1.1.163	20B	B.1.1.163	20B
IMB1-2/2021	B.1.1.163	20B	B.1.1.163	20B
IMB1-5/2021	B.1.1.163	20B	B.1	20A
IMB2-1/2021	B.1.1	20B	B.1.1.7	20I (Alpha, V1)
IMB2-2/2021	B.1.1.163	20B	B.1.1.163	20B

## Data Availability

Consensus FASTA files were deposited in the GISAID EpiCoV database (https://www.gisaid.org/, accessed on 12 May 2022) (Accessions are available in Appendix A Appendix A). Illumina consensus sequences for 5 resequenced samples were deposited in the GenBank (https://www.ncbi.nlm.nih.gov/genbank/, accessed on 12 May 2022) (accessions: MZ577122, MZ577123, MZ577124, MZ577125, and MZ577126). Nextstrain configuration files, the auspice JSON file, BEAST output logs, and trees files, and resulting log and tree files, as well as R scripts and data files used in the analyses, are available in Zenodo (https://doi.org/10.5281/zenodo.6406278, accessed on 12 May 2022) [53].

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
