# Peer review of "Molecular Analysis of SARS-CoV-2 Lineages in Armenia"

_viruses, 2022, doi:10.3390/v14051074_

Round 1

Reviewer 1 Report

The study investigates genomic epidemiology of SARS-CoV-2 in Armenia using appropriate sequencing and phylogenetic methods. The study provides a clearer picture and important information regarding SARS-CoV-2 lineages circulating in the country. However, major revisions are needed to improve the manuscript:

Introduction

  1. The authors mention in the introduction that: “Another set of samples from September-November 2020 was sequenced by our colleagues at the Vaccine and Infectious Disease Division, Fred Hutchinson Cancer Research Center (USA) which became available in May 2021”, however, the authors did not include how many samples or what the results of these genomes were, or where are they available. It is unclear from the introduction whether these were the genomes that the authors used for analyses in this study, or if these are separate (in which case more description should be provided).

Materials and methods

  1. The authors do not describe under what protocol the samples were collected and whether an IRB approval was obtained. An explanation should be added.
  2. Table 1 in materials and methods is unnecessary, I think that the text description is clear and already describes the data well.
  3. In section 2.4.4 Nanopore sequencing refers to Figure 1 that is supposed to show sequencing coverage but I do not see a figure representing this anywhere in the manuscript. Figure 1 in the manuscript shows daily incidence of SARS-CoV-2.
  4. In section 2.6 the authors mention that ” Phylogenetic analysis and strain reconstruction was performed using the SARS-CoV genomic epidemiology-specific pipeline implemented in Nextstrain”. It is unclear to me what is meant with strain reconstruction. Do the authors mean lineage identification, or lineage assignment?
  5. I wonder what the reason is that reference country-specific genomes were chosen randomly? Would it have not been better to run BLAST on Armenian genomes to find which ones in GISAID that were more closely related? With random selection of 10 genomes per country per month, and with the large amount of GISAID genomes (millions), chances are that the most related reference genomes will not be picked up by a random method. 
  6. For Bayesian analyses, were the ESS values considered and what was the threshold in that case?

Results and discussion

  1. The authors describe extensively in the Results the sources (countries) of SARS-CoV-2 introductions into Armenia. Are there any probability or likelihood values associated with these estimates that would give us an idea of how reliable these location origins are? For instance, BEAST location trait analysis provides probability values of each location. My concern with author’s country source conclusions is that the reference genomes were chosen randomly per country, and that most related genomes to the Armenian genomes were thus probably missed. This lack of more related data may give false relationships and may point to inaccurate sources of introduction. This limitation should further be discussed.
  2. Given that the authors extensively describe country sources in the results I find it curious that there is no figure showing the phylogenetic tree that supports these statements. Figure 2 represents (if I understand correctly) Bayesian tree of Armenian genomes only and their estimated TMRCAs and shows no geographical source results and no reference genomes. I think a Figure that supports country locations as well would be necessary.
  3. The authors state: “On the other hand, the deaths accompanying 21 (Delta) wave were considerably higher compared with the 21K (Omicron), which is in line with observations in other countries [40,41].” Authors show no data in the manuscript that support this statement for Armenia.
  4. The authors write: “Our results demonstrated that the majority of sequences harbor mutations in epitopes with high-binding affinity to protective HLA loci, while only a few sequences showed the presence of the mutations associated with low- binding HLA alleles” With close to 700 rows in table S4, this is not easily observed or clear from the supplemental table. Given that this is one of the main analyses of the manuscript, I would suggest the authors provide a summary table of this result in the main text of the manuscript, with exact numbers.
  5. The authors write: “The results of the analysis showed that 39 mutations in 143 samples spanned many regions targeted by primers and probes from different protocols (Supplementary Table S5). However, mutations targeting US CDC N3 and China CDC Orf1AB primers and probes did not influence the primer binding since we obtained the N gene PCR signal in all studied samples (Supplementary Table S1).” This section is unclear. What exactly were the findings, that 143 samples had mutations that did not match the primer? Or that primer-induced mutations were found in 143 samples? The authors should clarify. This is also not clear from looking at the Table S5. Furthermore, it is unclear what the authors mean with “mutations targeting US CDC N3 and China CDC Orf1AB primers”, how can mutations target primers? This needs to be clarified. I think this section needs to be described better.
  6. Figure 5. Frequency is denoted by a fraction number or a percentage. I would guess the Y axis here represents the number of missing sites instead? The authors should fix the units on the Y-axis.
  7. I think there is the possibility to calculate if there is a significant difference between Illumina and Nanopore in the number of missing sites, If this is significant, then one method is clearly superior in this than the other.
  8. Line 323 – sequencing type should be named consistently throughout the manuscript, here Illumina is referred to as short-read sequencing instead.
  9. The authors write: “We also have 5 samples sequenced by two methods, so we compare them to evaluate the correspondence of clade/lineage assignment with nanopore and short-read sequencing. 96%±6 (min-max: 83-99%) versus 93%±3 (min-max: 88-97%).” What do the numbers in this statement represent?
  10. Figure S2 is not labeled so it is unclear what represents Illumina and what represents nanopore consensuses, and why there are 2 different consensus sequences from Armenia.
  11. In Nanopore vs Illumina comparison, there are differences in Nextstrain, and Nextstrain+Pangolin calls. IMB2-1/2021 and IMB1-5/2021 are assigned differently by both tools when comparing Illumina and Nanopore. However, the authors do not discuss the discrepancy in the IMB2-2/2021 Nextstrain call. This would mean that the calls were wrong in about half of the examined samples. Seeing this data, I would not go ahead and state that Nanopore can be used as an efficient method for clade assignments. If anything, this would tell me that caution should be used when looking at Nanopore data, and that one must ensure only high quality data is being analyzed. This should be better discussed in the manuscript. In production of high quality data, user experience also matters.

Conclusions

  1. The authors state that “Interestingly, the majority of importations inferred by phylogeographic analyses were through airway travels” There is no evidence for this statement in the manuscript, and it is a speculation only.
  2. The authors write: “Our results also show a higher accumulation of mutations in regions covered by several primers/probes.” Higher than what?

Acknowledgments:

  1. The authors have clearly used GISAID reference genomes for their geographic and introduction analyses, and as they themselves state, 17,721 contextual sequences were used in the analyses. If these originate from GISAID, then the submitting and sequencing laboratories should be adequately acknowledged, as per GISAID requirements of a table mentioning all the involved laboratories.

Reviewer 2 Report

Reviewer’s report

Manuscript:

Molecular Genetic Analysis of SARS-CoV-2 Lineages in Armenia

In the current study, Diana Avetyan et al report the first molecular analysis of SARS-CoV-2 in Armenia during the two years of the COVID-19 pandemic. The authors studied several facets of SARS-CoV-2 strains circulating in Armenia: circulating lineages, comparison between Illumina and nanopore technologies, phylogeographic study and functional significance of mutations. Thus, the current study is an important addition to the global genomic surveillance literature.

I have the following comments that hopefully can help the authors in improving the manuscript.

  1. Title: please remove the word “genetic” from the title. Molecular analysis covers all the aspects you have described in the paper
  2. Keywords: please add Illumina and Armenia in the Keywords
  3. Line 24: you mention 6 clades and put only 5 please correct (20A is missing)
  4. Line 28-29: This sentence should be revised “The geographical diversity spanned countries from all continents”.
  5. Line 49: please delete the hyphen after the word country, and replace “and” by a coma
  6. Lines 48-52: very long sentence please reformulate it.
  7. Line 52: please remove for example just refer to references [9-12].
  8. Line 63: please add the GISAID accession numbers of the second set of sequences
  9. Line 63-65: Please reformulate the sentence
  10. Line 65: please replace Lab by laboratory
  11. Lines 68-72: This is the aim of your work; however, you are describing the principal results. I propose that you reformulate this paragraph. For example: “In the present study we report the first molecular analysis of SARS-CoV-2 virus in Armenia in order to (1) understand the emergence and the transmission of the virus, (2) identify the most prevalent lineages at different time points and (3) to investigate potential functional consequences of mutations detected in the sequenced Armenian samples”.
  12. Table 1 could be removed or transferred to supplementary material; the information are very well explained in the text no need for a table  
  13. Line 121: after the point do not use directly a number you should fully spell out “twelve” or N=12
  14. Line 128: You mention, “the coverage is presented in Figure 1” in Figure 1 we do not see any coverage, do you mean figure S1 (supplementary material?) Please edit this point.
  15. Line 171: you mention, “with parameters described in [9]”, please replace it by with previously described parameters [9].
  16. Line 177. You mention “The functional annotation of SARS-CoV-2 genomes from Armenia” you are speaking about the sequences performed within your study the word in Armenia is very large so here you replace “from Armenia” to “studied in the present study” or “from Armenia as of …”and here you put the date.
  17. Line 185: legend of figure 1, please add “confirmed” before cases
  18. Line 187: You mention “145 (75%) sequences met the quality criteria”, could you please specify the quality criteria you have used in the methods section
  19. Figure 3: it is not possible to distinguish between clades 19A, 20A and 20B all of them are represented with same Grey, please re-edit this figure using clear colors or annotate the clades directly on the branches
  20. Line 222: You mention, “These sequences were represented by PANGO lineages B.1 and B.1.1.” when the sequence is not good PANGO lineage could not specify the sub-lineage and it is generally B, B.1 or B.1.1 so I propose that this sentence should be deleted.
  21. Line 229: Are you sure here you are speaking about BA.1.1?
  22. Line 264: 6 and 1 please fully spell them or put a coma after them or n=6 and n=1,
  23. Figure 4 legend: specify that the calculation is performed on N=145 samples detected in Armenia
  24. Line 313. 86 directly after the dot please write it in letters or reformulate “out of 97, 86 were….
  25. Conclusion: the conclusion is very long and reports paragraphs that should be placed in discussion section please review the conclusion
